# An Exploration of Overconfidence and the Disposition Effect in the Stock Market

**Benomar Ikram [1,\*], Ben El Haj Fouad [1] and Chelh Sara [2]**

[1]  Interdisciplinary Research Laboratory in Economics, Finance and Management of Organizations, University Sidi Mohamed BEN Abdellah, Fes 30000, Morocco; fouad.belhaj@yahoo.fr
[2]  Research Laboratory in Entrepreneurship and Organizational Management, Fez Business School, Private University of Fez, Fes 30000, Morocco; chelh.sara@gmail.com
[\*]  Correspondence: ikrambenomar2016@gmail.com

**Abstract:** This paper offers a comprehensive empirical overview of the impact of overconfidence in the stock market, thus contributing to the existing research literature on this topic. The study employs a bibliometric approach that utilizes the VOSviewer to extract and analyze 277 articles registered between 1992 and January 2023. By providing a detailed analysis of the literature, this research expands our understanding of the impact of overconfidence in the stock market and offers avenues for future studies in this area. The results of this analysis are noteworthy, as they reveal several important findings. These include the exponential growth of scientific production in recent decades, the concentration of research in specific journals indexed in the Journal Citation Reports, the presence of institutional co-author networks, and the thematic and temporal segregation of financial behavior concepts. The most significant finding of this study is the identification of six major clusters: investor behavior during times of crisis; behavioral finance; herding and risk-taking concepts; psychological and cognitive decisions; emotions and decision-making; and the performance of stocks. This temporal evolution of research demonstrates the emergence of various perspectives on the relationship between individual financial behavior and the global market. This study represents a pioneering effort in the field of bibliometric analysis as it is the first to specifically examine the subject of overconfidence in the stock market using this method.

**Keywords:** overconfidence; stock market; behavioral finance; bibliometric analysis

## 1. Introduction

The behavioral tendencies of investors in financial markets have been extensively studied in the academic literature due to their potential to lead to suboptimal investment decisions. Two prominent biases studied in this area are overconfidence and the disposition effect. In the context of financial market efficiency, the Efficient Market Hypothesis (EMH), proposed by Fama in 1970 (Fama 1970), has been a widely debated and scrutinized theoretical framework. EMH assumes that financial markets are efficient when participants, who hold rational expectations, make decisions to maximize their expected utility. However, the validity of EMH has been challenged by empirical evidence and behavioral finance theories, which suggest that factors such as overconfidence among investors can impact financial market outcomes.

Overconfident investors are characterized by an overestimation of their own valuation abilities, as they overestimate the accuracy of the private information signals they possess (Daniel et al. 1998; Gervais and Odean 2001). Studies by those such as De Bondt and Thaler (1995), Odean (1998a, 1998b, 1999), and Gervais and Odean (2001) have shown that overconfident investors tend to trade more frequently and have contributed to the elevated trading volumes observed in financial markets.

Overconfidence can manifest in various forms, such as miscalculation (Lichtenstein et al. 1980; Yate 1990; Keren 1991; McClelland and Bolger 1994), overestimation of one's

abilities compared to others (Svenson 1981; Taylor and Brown 1988), illusion of control (Langer 1975; Presson and Benassi 1996), and unrealistic optimism (Weinstein 1980).

Measuring overconfidence in an investment decision-making context has been an ongoing challenge in the field, with no universally accepted method as of yet. In an effort to shed light on this phenomenon, researchers have formulated theories and generated testable implications based on two underlying hypotheses. The first hypothesis suggests that investors exhibit an overconfident perception of the accuracy of their personal information, whereas the second hypothesis posits that the extent of overconfidence is influenced by the self-attribution bias, and it depends on the outcomes that are obtained from the market (Deaves et al. 2008).

Deaves et al. (2008) utilized a calibration technique to advance the understanding of overconfidence and they developed a new model in this regard. Subsequently, Hoffmann and Post (2016) investigated this model further and concluded that investors who exhibit higher levels of confidence tend to hold stronger beliefs and they trade more frequently.

Building upon this research, several studies have attempted to explore the relationship between overconfidence and financial market outcomes in greater detail. For example, Tekçe and Yılmaz (2015) investigated the prevalence of overconfidence and its influence on investor behavior and returns. Talpsepp et al. (2014) conducted an experiment to determine the factors contributing to the disposition effect, which they found to be a combination of the following: a risky attitude towards losses, wishful thinking, and a misperception of price processes. Best and Grauer (2016) analyzed the use of prospect theory portfolios in asset allocation, taking into account limitations such as restrictions on short sales for risky assets, margin constraints, and the availability of risk-free lending and borrowing. Aspara and Hoffmann (2015) continued this line of inquiry by examining the role of personal responsibility factors in reducing an individual's susceptibility to the disposition effect.

An increasing number of scholars have been conducting and publishing articles on overconfidence in the stock market over the past two decades. Some scholars published systematic reviews on this topic. However, these literature reviews were focused on content analysis to discuss aspects related to the field of behavioral finance, rather than exploring the knowledge structure of the field through bibliographic and network analyses. This article seeks to explore the relationship between overconfidence and the disposition effect, and its potential impact on financial market outcomes, by adopting a bibliometric method to map important studies in the area. In this context, the research questions addressed in this study are:

RQ1: What is the distribution of overconfidence research in terms of the number of citations and publications per year, and research areas from 1992 to 2023?

RQ2: Who are the most influential authors, and which are the most influential countries, journals, and publications in the field of overconfidence research and stock market disposition effects?

RQ3: How have co-citation studies progressed, leading to meaningful clustering with a specific research focus?

RQ4: What are the most active areas, recent research trends, and emerging themes in overconfidence research?

To solve the questions above, this study conducted a thorough examination of the literature concerning overconfidence and disposition effects via a bibliometric analysis that utilized a VOSviewer. By systematically examining over 277 academic publications from the SCOPUS database, this method allowed us to visualize the knowledge structure and quickly understand the current state of the field of behavioral finance. The study includes an analysis of publication trends, types, top journals, and categories within the field. Additionally, citation analysis was performed to determine the most highly cited publications, and a reference co-citation analysis was conducted to identify potential future research opportunities by grouping and analyzing references. Currently, the use of quantitative-based tools for analyzing the literature has become prevalent, thus indicating their significance in terms of describing the gaps, the evolutionary trajectory, key trends,

and potential research directions of a topic in an objective manner, surpassing traditional descriptive reviews. Ultimately, this research aims to pinpoint any gaps and potential future avenues for investigation within the field of behavioral finance.

Despite a few literature reviews on specific topics of overconfidence over the past two decades, this study is among the first to use bibliometric methods to describe and analyze the evolution of the published literature on the topic. Indeed, although many studies focus on behavioral finance, there is a lack of comprehensive research analyzing the conceptual and intellectual foundations of financial knowledge. In that sense, this study represents the first comprehensive bibliometric review of financial education, thus providing practitioners, policymakers, educators, and academics with an up-to-date summary of the most recent advances in the field.

The rest of the article is structured as follows. The methodology section briefly discusses the history of bibliometric analysis and the methodology used in this study. We begin the results and discussion section by focusing on the distribution of the retrieved papers by publication year over time; after that, a range of analytical methods were employed, such as co-occurrence, bibliographic coupling, citation, co-authorship, and co-citation, in order to identify the articles, authors, countries, and organizations that have had the greatest influence in the field. The results are discussed progressively in this section. Section 4 discusses the current research directions and potential research opportunities based on the bibliometric analysis. Finally, the last section concludes this paper and presents its limitations.

## 2. Research Methodology

The purpose of this study was to conduct an in-depth analysis of the current research on overconfidence and behavioral biases in stock market investment decisions. To achieve this objective, we employed a bibliometric analysis methodology to review the existing literature comprehensively. In recent years, bibliometric analysis has become a widely used research method, drawing the attention of scholars. This method's growing popularity can be attributed to its capacity to process large quantities of data and its adaptability to various software applications, including Gephi and VOSviewer, and diverse information sources, such as Scopus and Web of Science. Researchers utilize bibliometric analyses to identify the prevailing trends in a particular field or journal, author and citation patterns, and to present an overview of the intellectual framework of a given area (Donthu et al. 2021).

The study consisted of a multiple-stage process, comprising a Literature Search, Data Collection and Pre-processing, a Bibliometric Analysis, Keyword Analysis, and Cluster and Content Analysis. By following this systematic approach, we aimed to provide a comprehensive and detailed analysis of the research subject, which would enable us to identify the current trends, patterns, and structure of the literature. This method allowed us to ensure that all the relevant literature was included and that the collected data were effectively analyzed and interpreted.

The figure below shows the major steps of the methodology adopted in the article, and each step will be developed later.

The initial stage of the study involved conducting a comprehensive literature search through the academic database, Scopus. This database was selected based on its reputation as the world's largest abstract and citation database of peer-reviewed literature, as well as its widespread usage among scholars worldwide (Parmentola et al. 2022). To perform the search, we identified a set of relevant keywords, which we applied to the keywords, titles, and abstracts of the papers. Our research methodology did not impose any restrictions on the types of documents included in the analysis, thus ensuring a comprehensive and unbiased review of the literature.

Search String: TITLE-ABS-KEY (overconfidence) AND TITLE-ABS-KEY (stock AND market).

The second stage of the study involved collecting and pre-processing the data. More specifically, we extracted a comprehensive set of data elements, including the title, authors,

subject area, document type, affiliation keywords, country, and publication year, which were then prepared for further analysis.

In the third stage of the study, a bibliometric analysis was conducted using the VOSviewer software, a commonly used tool for analyzing scholarly literature. The dataset used in the analysis consisted of 277 articles, which were analyzed to uncover various publication and citation patterns, influential articles, and author networks. This bibliometric analysis was used to identify research trends and to track the evolution of this field over time, as previously mentioned. The analysis also helped us to understand the distribution of research in terms of time and geographical location. By identifying the most productive authors and relevant documents, this analysis provided an overview of the most significant contributions to this field.

In addition, we conducted a keyword analysis using the same software to identify the main keywords used by authors in the literature on overconfidence and the stock market. This allowed us to map the most significant groups of keywords within the network.

Finally, a cluster analysis was employed to group similar keywords based on their co-occurrence patterns. The outcome of the cluster analysis, presented visually with the clusters heatmap, highlights the relationships between the keywords and the grouping of similar keywords. By utilizing cluster analysis, we identified the main themes in the literature and suggested future research directions based on these clusters. This was achieved by reading and analyzing the articles to identify the main concepts.

## 3. Results

In this results section, we aim to provide a comprehensive analysis of the current status of the field of overconfidence in the stock market. We will start by examining the overall scientific production in this area, including subject area analysis and a breakdown of contributions by country. We also explored the most relevant and impactful studies in this field. Furthermore, we conducted a citation analysis to identify the most influential studies and authors in the field of overconfidence in the stock market. Finally, we used keyword and cluster analysis to uncover emerging research trends and identify potential areas for further research.

### 3.1. Current Status of the Field

The topic in question has seen a substantial rise in interest, as evidenced by the proliferation of published research on the subject in recent years. This trend can be traced back to the initial publication on the topic in 1992, which served as a catalyst for subsequent investigations and studies. However, as is evident in Figure 1, the number of academic publications on this subject reached a peak in 2019. Indeed, according to the analysis, the highest percentage of published papers (69.54%) was observed during the most recent four-year period. The analysis of papers within the field revealed the following distribution of the total number of papers by year: 2022, 20.8%; 2021, 18.27%; 2020, 18.78%; and 2019, 11.67%. The field of behavioral finance, which concerns the impact of psychological and cognitive factors on financial decision-making, has gained popularity in recent years, and this could have contributed to the increased research on overconfidence and disposition effects in 2019. There are several factors that could explain the growing attention concerning overconfidence in the stock market. One of the main reasons may be the occurrence of significant economic events, such as market crashes or financial crises, that have brought attention to decision-making and market behavior. Changes in regulations and advances in research methods, such as behavioral finance, may have also facilitated the study of overconfidence and its effects.

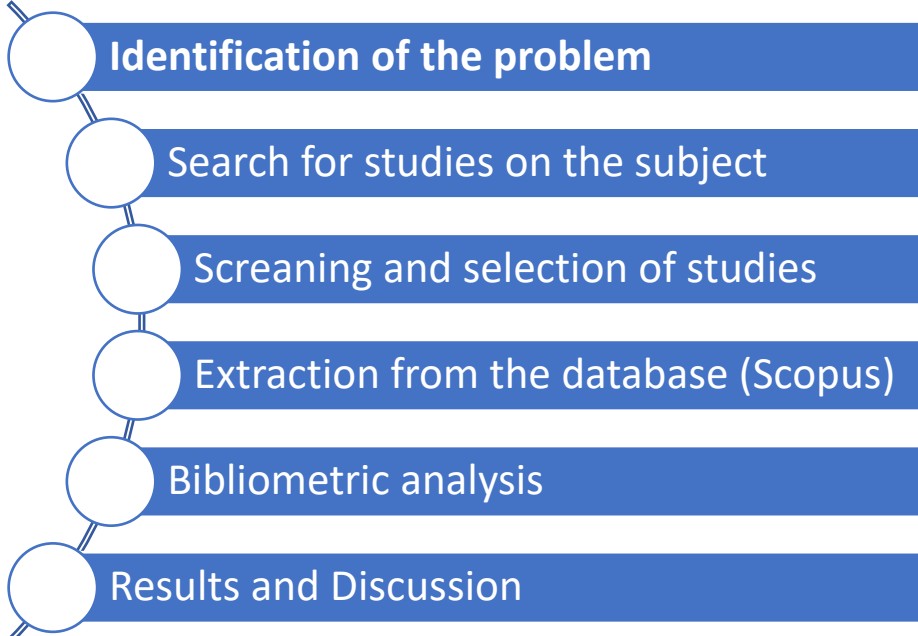

**Figure 1.** Methodology adopted in the research.

*3.2. Documents by Subject Area*

The topic of overconfidence in the stock market encompasses a diverse range of subject areas, as depicted in Figure 2. The analysis revealed that the majority of publications (40%) were categorized under the field of Business, Management, and Accounting, followed by Computer Science (6%), and Social Sciences (6%). The subject of psychology also contributed a significant number of publications (26%).

Finance researchers aim to comprehend how overconfidence can cause investors to make irrational decisions and its impact on market outcomes. Econometrics researchers focus on developing models to analyze the relationship between investor behavior and stock prices. Psychologists are keen to understand the cognitive biases that lead to overconfidence and ways to overcome them. Engineering and computer science researchers are exploring the development of algorithms and models that can help investors make better decisions amidst uncertainty. Finally, in management, studying overconfidence can aid organizations to comprehend the decision-making processes of their employees and executives, leading to overall better performances. Hence, it is evident that the subject of overconfidence in the stock market and investor behavior cuts across multiple disciplines and has implications that transcend beyond the business world.

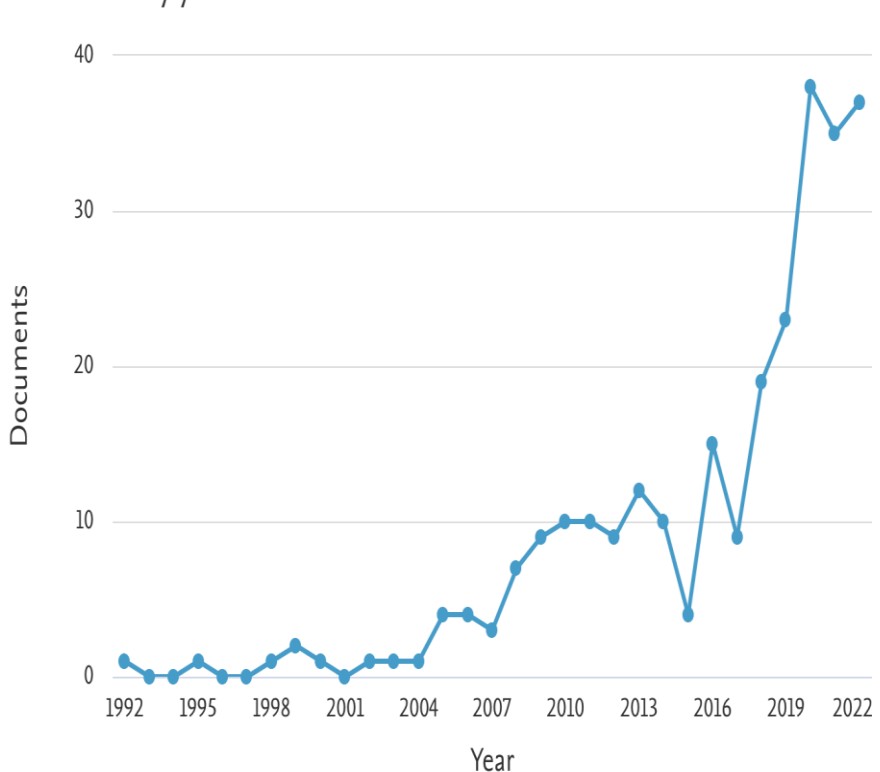

**Figure 2.** The annual number of research articles on overconfidence and the stock market indexed in Scopus from 1992 to January 2023.

### 3.3. Scientific Production by Country

To further explore the worldwide trend of overconfidence in the stock market, an investigation was carried out to find the level of interest in different countries (Figure 3). The map below presents the top 10 most productive countries in terms of research activities related to overconfidence and the stock market, revealing that the United States, China, and Taiwan contribute approximately 50% of the total worldwide publications. These countries are instrumental in terms of advancing research on overconfidence in the stock market. The United States is the most significant contributor, generating 57 publications in 132 journals, which accounts for 17.6% of the total worldwide publications. China closely follows as the second most productive country, with a total of 50 papers, followed by Taiwan with 29 papers. Notably, the United States is also the most productive country with regard to investor confidence, with 7513 total citations.

It can be challenging to determine the specific country or continent responsible for a given publication since many papers are the result of international cooperation, involving authors from multiple nations (Parmentola et al. 2022). A visualization produced by the VOSviewer in Figure 4 illustrates the distribution of countries or territories by area, with smaller distances indicating stronger connections between countries, thus suggesting robust intra-country collaboration. The figure displays co-authorship between 53 countries, grouped into 22 categories by color, with a total of 22 links and a combined link strength of 105. The United States is the most productive nation, with the highest number of links (34), followed by China (32 links), Taiwan (10 links), and others. International collaboration offers numerous advantages, such as networking, knowledge exchange, access to diverse expertise, and exposure to different cultures.

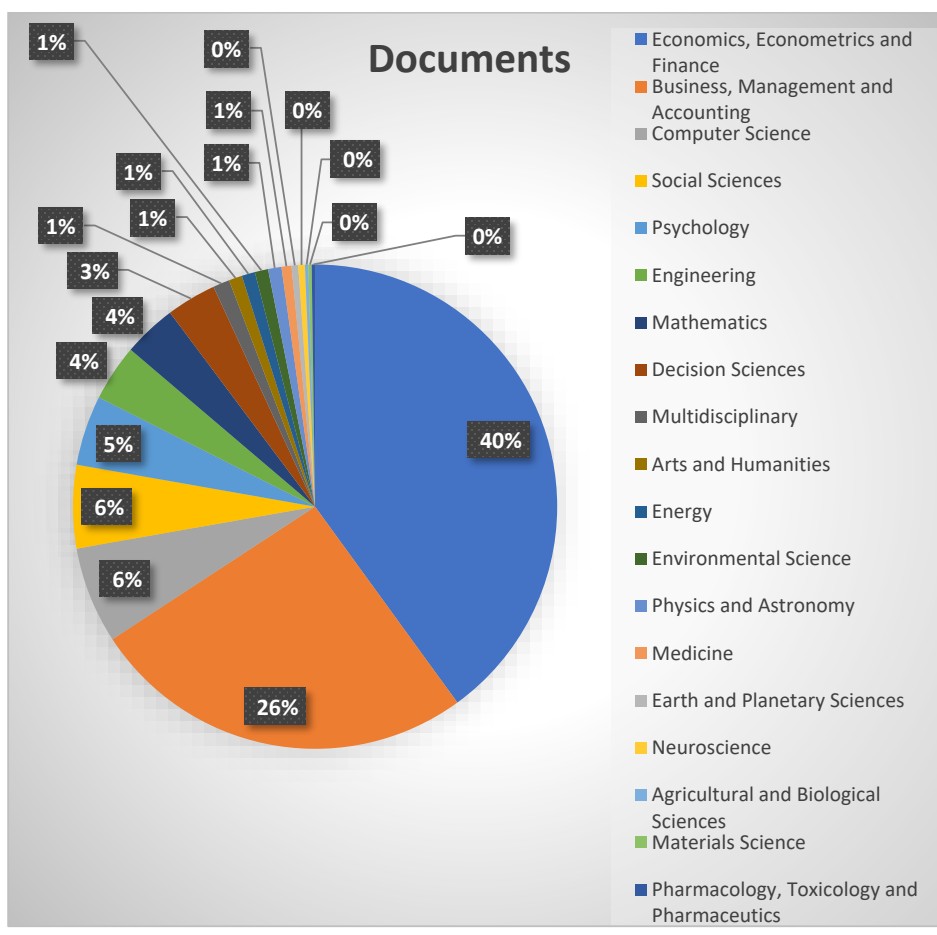

**Figure 3.** Research areas related to overconfidence in stock market (created with Excel).

# Documents by country/territory

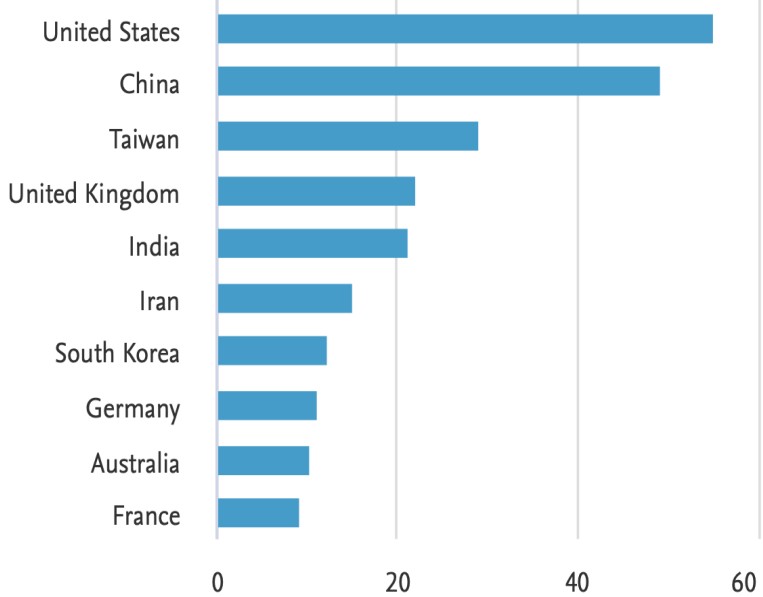

**Figure 4.** Top 10 most productive countries with regard to overconfidence and the stock market (created by the author using Google Maps).

*3.4. Most Relevant Contributions*

Table 1 presents an overview of the most prolific authors and their co-authors who have made substantial contributions to research on overconfidence and the stock market. Through their collaborative efforts, they have advanced the knowledge and understanding of these phenomena, resulting in significant progress in these fields. Notably, the table highlights that authors such as Daniel K., Barber B. M., Dunning, Lorenz J., and Statman have made noteworthy contributions to the literature on overconfidence, with Daniel K and Barber B. M. being the most cited authors, with citation counts of 2457 and 1463, respectively. Moreover, the close collaboration among the top 10 most published authors in recent years points to the growth and maturity of research in the field of overconfidence and the stock market.

**Table 1.** The top 10 most prolific authors.

| Rank | Document | Years | Citations | Links | The Most Cited Article (Reference) | Journal |
|---|---|---|---|---|---|---|
| 1 | Daniel K. | 1998 | 2457 | 94 | Investor psychology and security market under- and overreactions. | Journal of Finance |
| 2 | Barber B. M. | 2000 | 1463 | 55 | Trading is hazardous to your wealth: the common stock investment performance of individual investors. | Journal of Finance |
| 3 | Dunning D. | 2004 | 1187 | 1 | Flawed self-assessment implications for health, education, and the workplace. | Psychological science in the public interest, supplement |
| 4 | Lorenz J. | 2011 | 571 | 0 | How social influence can undermine the wisdom of crowd effect. | Proceedings of the National Academy of science of the United States of America |
| 5 | Statman M. | 2006 | 409 | 59 | Investor overconfidence and trading volume. | Review of Financial Studies |
| 6 | Chen G. | 2007 | 258 | 17 | Trading performance, disposition effect, overconfidence, representatives bias, and experience of emerging market investors. | Journal of behavioral decision-making |
| 7 | Billett M. T. | 2008 | 253 | 7 | Are overconfident CEOs born or made? Evidence of self-attribution bias from frequent acquirers. | Management Science |
| 8 | Daniel K. | 1999 | 168 | 8 | Market efficiency in an irrational world. | Financial Analysts Journal |
| 9 | Shiller R. J. | 1999 | 140 | 2 | Chapter 20: Human behavior and the efficiency of the financial system. | Handbook of Macroeconomics |
| 10 | Nosi'c A. | 2010 | 132 | 3 | How riskily do I invest? The role of risk attitudes, risk perceptions, and overconfidence. | Decision Analysis |

*3.5. Citation Network Analysis*

When evaluating an author's impact in a specific field, it is crucial to consider various metrics beyond the number of publications. Citations serve as another significant metric that sheds light on an author's influence in the field. The network presented in Figure 5 demonstrates a group of authors who have been cited more than ten times in the area of overconfidence in the stock market. The central position of the most cited author, Daniel K., in the network illustrates his prominent influence in the field. The network can be divided into groups based on the level of collaboration between authors, which are represented by various colors. The size of the circles indicates the frequency of occurrence of each author, and the lines between the circles denote collaborative relationships.

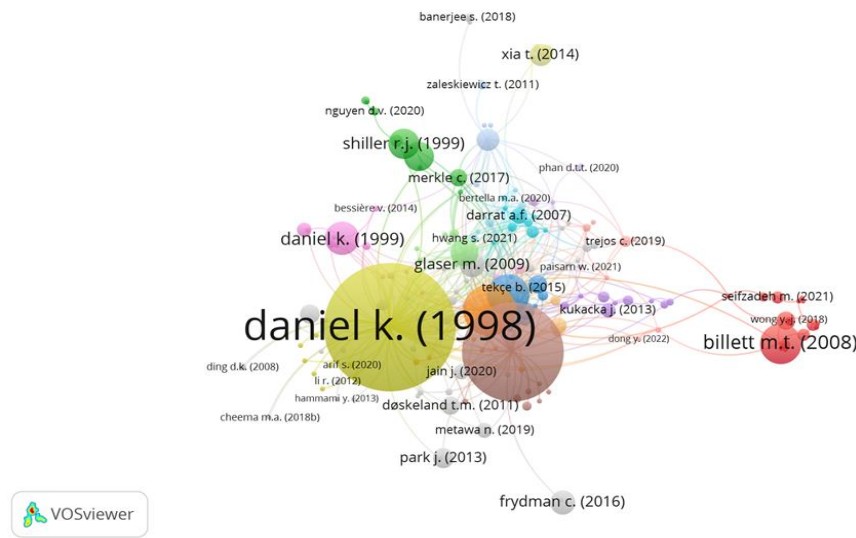

**Figure 5.** Co-citation authors network (created using the VOSviewer).

### 3.6. Contribution by Journals

Table 2 provides clear evidence that publications in the fields of finance and research, with regard to overconfidence and the stock market, appear frequently in journals related to finance and psychological sciences, as well as in management sciences and decision analysis. This observation aligns with the findings from the prior analysis of publication categories. These findings suggest that the overlap between these disciplines is an essential aspect of advancing knowledge in this field. Notably, the significant presence of publications in management sciences and decision analysis also underscores the multidisciplinary nature of this area of study. Such observations reflect the growing trend of interdisciplinary research, thus emphasizing the importance of collaboration between scholars from different fields. Therefore, the results from Table 2 underscore the need for a collaborative approach to research on overconfidence and the stock market that encompasses insights from finance, psychology, management sciences, and decision analysis. By embracing a collaborative approach, scholars can enhance the development of theory and practice in this area and contribute to the growth of interdisciplinary research. The most prolific publisher, was the American Finance Association (AFA), followed by the Association for Psychological Science, National Academy of Sciences, Society for Financial Studies (SFS), and John Wiley & Sons; these comprised the top five publishers.

**Table 2.** Top ten journals cited in research on overconfidence.

| Source | Publisher | TP * | TC ** | SJR *** | SNIP **** | Cite Score |
|---|---|---|---|---|---|---|
| Journal of Finance | American Finance Association (AFA) | 5.829 (2020) | 50,844 (2020) | 3.827 (2020) | 2.977 (2020) | 6.7 (2019) |
| Psychological science in the public interest, supplement | Association for Psychological Science (APS) | 3.838 (2019) | 5087 (2019) | 0.813 (2019) | 1.936 (2019) | 3.7 (2019) |
| Proceedings of the National Academy of science of the United States of America | National Academy of Sciences (NAS) | 9.58 (2020) | 891,906 (2020) | 10.637 (2020) | 8.986 (2020) | 14.5 (2019) |
| Review of Financial Studies | Society for Financial Studies (SFS) | 7.936 (2020) | 45,919 (2020) | 7.988 (2020) | 6.852 (2020) | 9.9 (2019) |

**Table 2.** *Cont.*

| Source | Publisher | TP * | TC ** | SJR *** | SNIP **** | Cite Score |
|---|---|---|---|---|---|---|
| Journal of behavioral decision-making | John Wiley & Sons | 2.648 (2020) | 4928 (2020) | 1.924 (2020) | 1.954 (2020) | 3.1 (2019) |
| Management Science | INFORMS (Institute for Operations Research and the Management Sciences) | 7.098 (2020) | 57,938 (2020) | 4.841 (2020) | 4.948 (2020) | 6.5 (2019) |
| Financial Analysts Journal | CFA Institute | 2.726 (2020) | 7865 (2020) | 0.959 (2020) | 2.066 (2020) | 3.3 (2019) |
| Decision Analysis | INFORMS (Institute for Operations Research and the Management Sciences) | 3.096 (2020) | 7973 (2020) | 2.634 (2020) | 2.744 (2020) | 3.8 (2019) |

*: Total publication. **: Total citation. ***: Scientific journal ranking. ****: Source normalized impact per paper: measures contextual citation impact by weighting citations based on the total number of citations in a subject field.

### 3.7. Analyzing Research Trends on Overconfidence in the Stock Market

In this section, our aim is to analyze the research trends on overconfidence in the stock market. To achieve this goal, we will use a two-step approach, starting with a keyword analysis followed by a cluster analysis. The keyword analysis will help us to identify the most common keywords and phrases used in the literature on overconfidence in the stock market. This will give us a broad understanding of the research trends in this area. The cluster analysis, on the other hand, will allow us to group together the articles based on similarities in keywords, topics, and other characteristics.

Keyword occurrence analysis is a valuable tool used to evaluate the frequency of specific terms in a given set of documents, such as research articles, books, and reports. In bibliometrics and scientometrics, keyword co-occurrence analysis is a popular technique used to determine the most commonly associated and relevant terms in a particular research field, as depicted in Figure 6 (Ellegaard and Wallin 2015). Both types of analysis can help to identify critical trends and themes in a particular discipline.

To identify the most important areas of overconfidence in stock market, 1185 keywords were analyzed. The co-occurrence map was generated by considering the keywords that appeared at least three times in all the collected documents. A total of 62 keywords met this threshold and were interconnected by 426 links. The present study utilized the "temporal display" feature in the VOSviewer, which is a powerful tool for visualizing the scholarly literature on overconfidence in the stock market. This feature facilitates the identification of emerging topics and offers a comprehensive means of analyzing the field in a nuanced and sophisticated manner.

As is evident in Figure 6, the term "Overconfidence" was the most frequently encountered keyword, with a total of 127 instances identified, along with 173 links to other keywords. Moreover, the term "Behavioral finance" was observed 32 times and was found to have 69 links to other keywords. It is not surprising that the keywords "Overconfidence" and "Behavioral finance" were among the most frequently encountered terms in a search related to overconfidence in the stock market. These terms reflect the widespread recognition of the influence of behavioral biases on financial decision-making and the importance of understanding these biases in order to make more informed and rational investment decisions.

The figure above not only displays the most frequently utilized keywords in the literature related to behavioral finance, but it also highlights the emerging topics in this field. To understand the topics that have attracted researchers and practitioners in behavioral finance over the last ten years (from 2012 to 2022), we analyzed the temporal patterns in the literature, and as a result, we found that the keywords, "COVID-19", "China", "loss aversion", "investor attention", and "decision making", with regard to stock price crash risks (green and yellow), have been most frequently employed during the last two years, as demonstrated in the figure. The keywords are indeed related in the context of the stock market, and their emergence over the past two years can be attributed to several factors. Firstly, the outbreak of the COVID-19 pandemic in early 2020 had a significant impact on the global financial markets, leading to increased volatility and uncertainty. This sudden and dramatic shift in market conditions likely prompted many investors to become more risk averse, leading to greater attention on the concept of "loss aversion" during decision-making processes. Secondly, the pandemic has also led to significant changes in the way that investors engage with the stock market. With many people forced to work from home, and with limited opportunities for social interaction, the level of "investor attention" has increased as individuals have more time to devote to monitoring and analyzing market trends. Finally, the rapid pace of change and uncertainty brought about by the pandemic has highlighted the importance of effective decision-making in the stock market. This, in turn, has led to increased interest in the various psychological and behavioral factors that influence decision-making, such as "loss aversion" and "investor attention" (Shrotryia and Kalra (2023)).

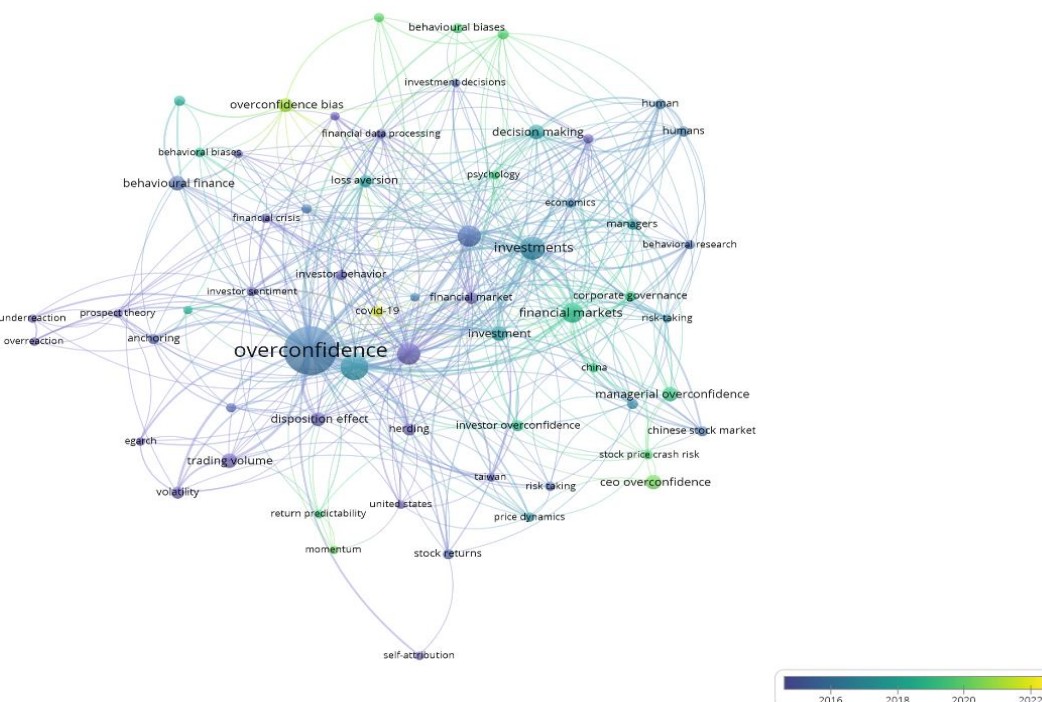

**Figure 6.** Keyword co-occurrence network.

Cluster analysis:

In our analysis, we identified six distinct clusters of co-occurring keywords (Table 3) and (Figure 3).

**Table 3.** Authors' keywords co-occurrence network.

| Cluster | Color | 62 Items | Main Items |
|---|---|---|---|
| 1 | Red | 13 Items | CEO overconfidence, China, Chinese stock market, corporate governance, COVID-19, financial markets, investor overconfidence, managerial overconfidence, managers, stock price, stock price crash risk. |
| 2 | Green | 12 Items | Anchoring, behavioral finance, disposition effect, EGARCH, market efficiency, overconfidence, overreaction, prospect theory, trading volume, underreaction, volatility. |
| 3 | Blue | 11 Items | Financial market, herding, investment, momentum, price dynamics, return predictability, risk taking, stock market. |
| 4 | Yellow | 11 Items | Behavioral biases, behavioral finance, emerging markets, financial literacy, herding behavior, individual investors, investor behavior, overconfidence bias, psychology, self-attribution bias. |
| 5 | Purple | 9 Items | Behavioral biases, behavioral research, commerce, financial crisis, financial data processing, investment decisions, investor psychology, investor sentiment, loss aversion. |
| 6 | White | 6 Items | Self-attribution, stock return. |

Figure 7 display the various clusters that were identified, each represented by a distinct color as indicated in Table 2. A label was assigned to each cluster to provide a concise summary of the main topics discussed within them. We will now proceed to elaborate on the main ideas contained in each of these clusters.

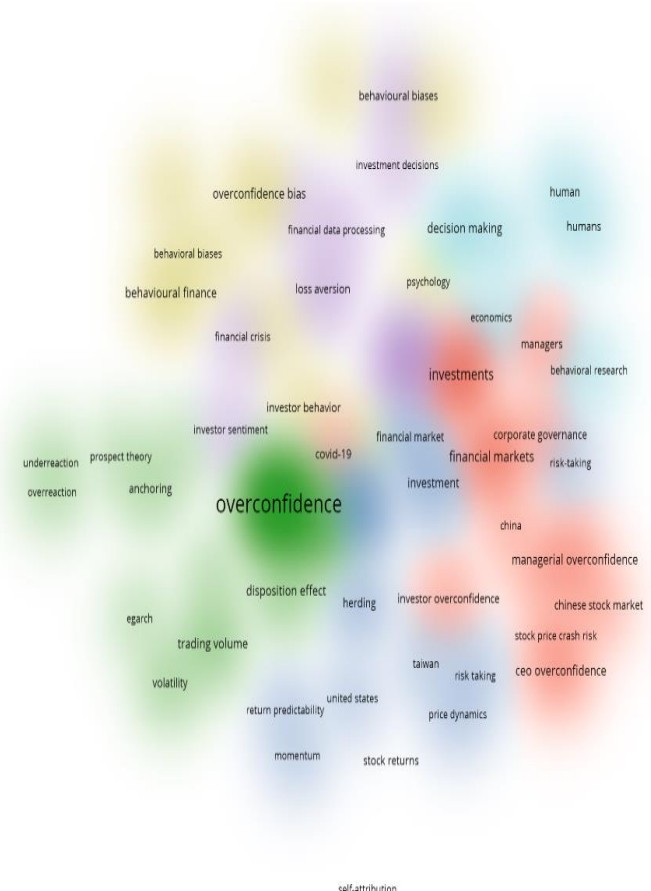

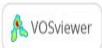

**Figure 7.** Cluster density visualization.

Cluster 1:

Overconfidence, Risk, and Crisis Impact on the Stock Market.

Financial markets play a crucial role in the economy as they efficiently allocate capital to companies and projects with the highest expected returns, and they provide liquidity to facilitate transactions. However, financial markets are also subject to speculative bubbles, volatility, and systemic risk, as evidenced by the global financial crisis of 2008 that was precipitated by the collapse of the US housing market. In 1973, Malkiel (Malkiel 1973) published his seminal book, "A Random Walk Down Wall Street," which presents the argument that stock prices follow a "random walk", and that it is impossible to consistently outperform the market using technical or fundamental analyses. Malkiel's work has been influential in the field of financial economics, as it challenges the notion that individual investors or professional money managers can consistently outperform the market. The book's argument supports the theory of efficient markets, which posits that asset prices always fully reflect all available information, and that active management is unlikely to generate higher returns than a passive approach that mirrors the market.

In addition to Malkiel's argument, Taleb's work in "The Black Swan" emphasizes the importance of understanding and preparing for rare and unpredictable events, such as the attacks that took place on 11 September and the 2008 financial crisis, that can have profound impacts on financial markets and the economy. This idea is particularly relevant in the context of the COVID-19 pandemic, which has had an unprecedented impact on global markets, and which highlights the need for robustness and resilience in the face of unexpected events.

The research in this cluster calls for a better understanding of the role of overconfidence in the financial markets to develop effective strategies for reducing stock price risk and market volatility.

Cluster 2:

Behavioral Finance Phenomena in Market Analyses.

Behavioral finance is an interdisciplinary field that integrates psychology and economics to better understand financial decision-making. This approach posits that decisions are not solely driven by rational economic factors, but are significantly influenced by psychological and emotional factors such as cognitive biases and emotions. By incorporating psychological insights into conventional economic theories, a more comprehensive understanding of financial decision-making can be gained.

Several key concepts in the field of behavioral finance include the disposition effect, the framing effect, and anchoring. The disposition effect refers to the tendency to hold onto losing investments while selling winning ones; this can be attributed to the emotional pain associated with a realized loss. The framing effect highlights how the presentation of information can influence decision-making, and investors may take different risks depending on how financial information is presented to them. Anchoring is a cognitive bias wherein individuals rely too heavily on the first piece of information they receive, leading to skewed estimates or judgments.

Market volatility, which is the degree of variation or fluctuation in asset prices, can be influenced by psychological biases and the emotional responses of investors. This can result in herding behavior, overreaction to news and events, and other forms of irrational behavior that can increase volatility. The field of behavioral finance has significantly contributed to our understanding of financial decision-making, shedding light on the role of psychological factors in shaping these decisions. (Hersh Shefrin 2001; Nofsinger 2010; Thaler and Ganser 2015).

Cluster 3:

Financial Markets and Investment Dynamics.

Herding and risk-taking are two important dynamics that influence financial markets and investment decisions. Herding behavior refers to the tendency of investors to follow the actions of others in the market, rather than making independent decisions based on their own analysis of market fundamentals. This behavior can lead to market bubbles

and crashes, as investors enter or exit a particular asset at the same time, creating a self-reinforcing cycle that amplifies market volatility. Risk-taking, on the other hand, refers to the willingness of investors to take on risk in the pursuit of higher returns. This behavior can result in significant gains or losses, depending on the success of the investment strategy. Financial markets are shaped by the interplay between these two dynamics, as investors seek to balance the desire for high returns with the need to manage risk and avoid the negative effects of herding behavior. Understanding these dynamics is essential for investors to make informed decisions and achieve their investment goals. Sabir et al. (2019). Kumar and Goyal (2016).

Investment decisions refer to the choices investors make concerning where to allocate their capital. These decisions may include the purchase of stocks, bonds, real estate, or other assets in the hope of earning a return. When making investment decisions, investors generally consider a variety of factors, such as the potential return on the investment, the level of risk involved, the personal financial situation, and the investor's objectives.

To manage these dynamics, Markowitz introduced modern portfolio theory in 1952 (Markowitz 1952). This theory provides a mathematical framework for selecting a portfolio of assets that maximizes the expected return for a given level of risk or minimizes the risk for a given level of expected return. Markowitz emphasizes that international diversification is crucial for reducing overall portfolio risk, as it allows investors to spread their investments across different economic environments and political regimes. However, despite the potential benefits of international diversification, French and Poterba (1990) identified the home bias puzzle, wherein domestic investors tend to have a higher proportion of their portfolios invested in domestic assets than expected based on the relative size of global markets. This suggests the influence of behavioral factors on investment decisions, such as familiarity bias, information asymmetry, and regulatory and tax considerations. Understanding these factors is essential for investors seeking to optimize their portfolios and achieve their investment goals.

Cluster 4:

Psychological Biases when Investing in the Stock Market.

The behavior of stock markets refers to the fluctuations in the prices of shares traded on stock exchanges, which can be influenced by a multitude of factors such as economic conditions, company performance, and investor sentiment. Economic conditions, such as interest rates, inflation, and gross domestic product (GDP), are known to have a significant impact on share prices, with strong economies usually exhibiting good company performances, which subsequently leads to increases in share prices. Conversely, weak economies tend to negatively affect company performances and drive down share prices. Investor sentiment, which represents the general mood of investors, is also a critical factor influencing stock market behavior. Optimistic investors tend to buy shares, driving up prices, whereas pessimistic investors tend to sell shares, leading to price declines.

Indeed, psychological biases are inherent in human decision-making, and they have a significant impact on investor behavior in stock markets. Overconfidence bias, where investors overestimate their ability to predict market trends, can lead to excessive risk-taking and poor portfolio diversification. Self-attribution bias, where investors attribute their successes to their abilities and their failures to external factors, can prevent them from learning from their past mistakes, and thus, they may continue to make poor investment decisions (Barber and Odean 2000; Shefrin 2002; Fama and French 1988; Baker and Wurgler 2007; Aljifri 2023).

Financial literacy is a crucial aspect of personal finance that helps individuals make informed financial decisions. One critical concept in behavioral finance is cognitive biases, which refers to systematic errors in thinking that lead to irrational financial decisions. By recognizing these biases, individuals can better evaluate their financial decisions and improve their overall financial literacy. Another important concept in behavioral finance is the "mental accounting" approach, which highlights how people categorize and think about their financial resources. This approach can affect their spending and saving decisions, as

individuals may prioritize certain expenses based on how they mentally account for their finances. When aware of mental accounting biases, individuals can make more informed financial decisions and better allocate their financial resources.

Cluster 5:

Unpacking the Emotional Toll of Loss Aversion on Stock Market Investors.

In addition to cognitive biases and overconfidence, emotions are also important in shaping investment decisions. Fear and greed, for example, can cause investors to overreact to market news, leading to suboptimal investment outcomes. Moreover, research has shown that personality traits such as risk tolerance, optimism, and impulsivity can affect investment decisions. For example, highly impulsive individuals may be more likely to engage in risky investments. Social influences are also important, with studies showing that investors are influenced by the behavior of others, leading to herding behavior and market bubbles.

To further understand the role of emotions in financial decision-making, researchers have suggested the need to develop more accurate models of investor behavior that incorporate both cognitive and emotional factors. Additionally, there is a growing interest in the cultural aspects of investment decision-making, as cultural differences may impact the way in which investors perceive and respond to risk and uncertainty. For example, studies have shown that in collectivist cultures, people may be more risk-averse and place greater importance on social norms and relationships, whereas in individualistic cultures, people may be more willing to take risks and prioritize personal achievement. Further research is needed to better understand these cultural differences and their impact on decision-making regarding investments. Ultimately, a better understanding of the complex interplay between cognitive, emotional, and cultural factors can lead to the development of more effective investment strategies and financial products that better meet the needs and goals of investors (Baker 2009).

Loss aversion is a psychological phenomenon wherein individuals experience a stronger emotional reaction to losses than to equivalent gains. This phenomenon has important implications for stock market investments, as investors may be more likely to hold on to losing investments in the hope of recouping their losses, rather than selling and realizing a loss. This behavior can be driven by emotions such as fear and regret, which can cloud rational decision-making and lead to poor investment outcomes. Research concerning behavioral finance has shown that loss aversion is one of the key drivers of investment behavior, as investors often hold on to losing stocks for longer periods than they should in an attempt to avoid the pain associated with losses. This can result in missed opportunities for gains, as well as higher levels of portfolio risk. Understanding the link between loss aversion and emotions is critical for investors to make informed investment decisions and manage their portfolios effectively. By acknowledging the role that emotions play in investment decisions, investors can take steps to mitigate the impact of loss aversion and other biases, such as diversification, risk management strategies, and regular reviews of investment portfolios.

Cluster 6:

Self-Attribution Bias and Stock Return: How Overconfidence Impacts Investment Performance.

Stock performance is a key metric that is used to evaluate the value and returns of an investor's portfolio. It refers to the change in the value of a stock over a given period of time, and it can be measured using a variety of techniques. One widely used measure of stock performance is stock return, which represents the gain or loss made by an investor holding the stock over a given period.

Several factors can affect the performance of a stock. Economic conditions, such as interest rates, inflation, and gross domestic product (GDP), can have a significant impact on the stock market and individual stocks. Additionally, company-specific factors, such as financial performance, management, and growth prospects, can affect the performance of a

stock. Finally, market sentiment, which reflects the general investor sentiment towards the stock market, can also influence the performance of individual stocks.

Moreover, self-attribution bias, a cognitive bias in which individuals attribute their successes to their own abilities and efforts, while attributing their failures to external factors beyond their control, can also impact stock performance. This bias can lead to overconfidence and suboptimal investment decisions, which can have a negative effect on the performance of a stock. Investors who are affected by self-attribution bias may be more likely to hold on to losing stocks for too long, or they might try to emulate past performances, thus resulting in missed opportunities and losses. On the other hand, the bias may also result in being overly optimistic and it may cause there to be a higher demand for a particular stock, which can drive up its price (Ngene and Mungai (2022)).

## 4. Discussion of Findings and Future Research Directions

The bibliometric approach used in this study of overconfidence in stock markets revealed several interesting findings. First, it was found that overconfidence has been a topic of interest in the field of finance and economics for several decades, with a steady increase in the number of publications on the subject in recent years. Second, the study identified several key themes in the literature, including the impact of overconfidence on trading behavior, the relationship between overconfidence and risk-taking, and the role of overconfidence in market bubbles and crashes. Third, the analysis revealed that certain research methods, such as experimental studies and surveys, have been used to investigate overconfidence more often than others.

Although there has been some research on the role of overconfidence in stock market investment, there are still several gaps in our understanding of this phenomenon. For example, more research is needed to explore the mechanisms through which overconfidence affects investment decisions, such as whether it leads to a preference for certain types of investments or a tendency to hold on to losing positions for too long. Additionally, studies are needed to examine the impact of overconfidence on investment outcomes in the long term, as well as in different market conditions. Another area that requires further investigation is the potential interaction between overconfidence and other behavioral biases, such as herding or anchoring, and how this affects investment decisions. There is a need for more research on interventions that can help mitigate the negative effects of overconfidence on investment performance, such as education or feedback mechanisms. Additionally, more research could be conducted to understand the impact of cultural factors on overconfidence and investment decisions.

Overall, the bibliometric analysis provides valuable insights into the state of research on overconfidence in stock markets, and it highlights potential areas for future research. By continuing to build on these findings, researchers can deepen their understanding of the complex dynamics at play in financial decision-making, and they can work towards improving the accuracy of investment predictions and outcomes.

## 5. Conclusions

The relationship between overconfidence and the disposition effect in the stock market has been widely researched and discussed in the academic literature. Studies have found that overconfidence is a significant contributor to the disposition effect and can lead to poor investment decisions and decreased returns. In this context, we are examining the same subject matter but with a more holistic approach. As a matter of fact, our study provides an in-depth analysis of the current state of research on overconfidence and its effects on the financial market; this was achieved by analyzing 277 publications extracted from the Scopus database. The VOSviewer, a software commonly utilized in the domain of bibliometrics, has been utilized to perform network and visualization analyses with the objective of identifying key journals and emerging themes and concepts within the field. The study provides an original perspective as it presents a comprehensive analysis of the publication trends over time and it identifies key gaps in the literature.

The outcome of this quantitative bibliometric analysis can be described in the following manner: the increase in the number of publications from various authors suggests that international collaboration may be a contributing factor in stimulating the pace of scientific publications. The number of articles published since 2019 has significantly increased, with the majority concentrated in the field of business, management, and accounting. Our bibliometric study indicates that authors affiliated with the United States were the most productive, with a total of 57 publications. China follows closely behind with 50 publications, thus demonstrating significant involvement in this field of research. The analysis of keywords resulted in six distinct clusters related to the subject. The first cluster deals with economic crises, the second with behavioral finance, and the third with herding and risk-taking concepts. The fourth cluster explores psychological and cognitive factors, the fifth addresses the links between emotions and decision-making, and the sixth focuses on self-indulgence bias and its impact on investor performance. These classifications help to better understand the diversity of research carried out on the overconfidence bias of investors in the stock market, and it offers avenues for future studies on the subject.

It is important to acknowledge the limitations of the current study which must be addressed in future research. First, the use of the Scopus database only may have resulted in the exclusion of relevant articles, which could be remedied by including additional sources such as Google Scholar, Web of Science, or ScienceDirect, as well as dissertations and theses. Although a bibliometric analysis provides valuable insights into the field of food industry supply chain resilience, it cannot fully explain the reasons behind the results. Therefore, future research could incorporate social science methods, such as expert interviews, to gain a deeper understanding of underlying factors. Additionally, the selection of a limited number of nodes in the VOSviewer co-occurrence analysis may have excluded important publications, thus suggesting that alternative bibliometric software could be utilized in future studies.

As a perspective for future research, we suggest conducting a meta-analysis, which is a statistical technique widely used for literature reviews. Meta-analyses involve combining the results of multiple studies to generate a summary effect size, and it is commonly employed in quantitative research to synthesize findings and draw generalizable conclusions. Meta-analyses can be particularly useful when there are conflicting or inconsistent results in the literature. It can help identify sources of variation and provide a more precise estimate of the effect size. This rigorous and objective technique can provide a quantitative estimate of the effect size, but it is limited by the quality and availability of the underlying data (Haidich 2010).

**Author Contributions:** Conceptualization, B.I. and C.S and B.E.H.F.; methodology, B.I. and B.E.H.F.; software; validation B.I. and C.S.; writing—original draft preparation, B.I.; writing—review and editing, B.E.H.F.; visualization, B.E.H.F.; supervision, B.E.H.F.; project administration; funding acquisition, B.I. All authors have read and agreed to the published version of the manuscript.

**Funding:** This research received no external funding.

**Informed Consent Statement:** Not applicable.

**Data Availability Statement:** Not applicable.

**Conflicts of Interest:** The authors declare no conflict of interest.

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
