# Peer review of "An Exploration of Overconfidence and the Disposition Effect in the Stock Market"

_ijfs, doi:10.3390/ijfs11020078_

Round 1
Reviewer 1 Report
1. X-axis of figure 1 has a date of 2025. If the year is the frequency, then the author should report it till 2022.
2. Why there is a sudden drop at the end of figure 1? If it is due to the first quarter of 2023, then it should be scaled accordingly to reflect an appropriate trend.
3. What do we learn from section 3.2 and figure 2.? It is not clear.
4. Figure 3 is a bit misleading and country names are not clear. Authors can simply use a bar diagram or a table that ranks countries.
5. What new observations or insights do we get from figure 4? The author should explain.
6. In fact, all the tables and figures just provide some statistics that could be generated from the software. For to make it a research paper, authors should bring insights, and inferences with appropriate economic intuitions. For now, the present work looks methodologically driven experiment on a select area.
7. Authors should make a proper table to highlight unexplored areas under the broad theme selected to give proper recommendations.
8. They should also explain other methodologies like meta-analysis, and morphological-analysis of doing a literature review and have a discussion on how this methodology contributes.
Reviewer 2 Report
Dear Authors,
Below you can find my review report for your article "An Exploration of Overconfidence and the Disposition Effect in the Stock Market: A Bibliometric Review" sent to Interntional Journal of Financials Studies.
The Abstract is well-written and it doesn't need any adjustments.
In chapter "2. Research methodology" you present the steps of the research methodology. I suggest you to include figure with the steps, because a visual image will be more useful to the readers.
Starting with the line, the alignment of the new paragraphs seems to be different from the previous lines. Please revise this aspect and use the same justification for alignment in the text.
I'm not very sure, but it seems you are using a different font type begining with the line 370. Please take a look and re-format the text if necessary.
The section "3. Results" and "3.1 Current status of the field" starts without any descriptive text. Please avoid this approach. Each new (sub)section should start with a short introductice sentence, so that the readers understand the context of that section.
The Figure 1 is entitled "The annual and cumulative number of research articles on overconfidence and the stock market indexed in Scopus from 1992 to January 2023." As I can say in the image, you presented just the annual number of the research articles. I couldn't see the cummulative number. Please correct this situation, because the title of the figure doesn't fit the image.
The references are pretty-old and you should add some "fresh" references. I recommend you to include the following useful resources in your article: https://doi.org/10.3390/math8010120, http://www.ecoforumjournal.ro/index.php/eco/article/view/884, https://doi.org/10.3390/joitmc6030071, https://doi.org/10.3390/jrfm15080319. These resources will bring a fresh approach in your article.
At the line 205, you have a sentence starting with "As expected, ....". Please avoid using this syntagm, because you are conducting a research. This expressions ("as expected") suggests a data manipulation, but we are discussing about a scientific research.
The "Figure 2 : Research areas on overconfidence in stock market (created with Excel)" is wrong. It is a pie-chart and the sum of the "slices" should be 100%. But in your figure, you have slices with 180% and 116%. It is totally wrong. Please pay attention to this issue and correct it.
At line 274, the title of the figure 5 should be reformatted. The font size is unusual.
The table starting at line 358 must have a number and a title.
It is not very clear the methodology used to discover the clusters. Please include a detailed description of it, so that the readers understand the entire context of your research.
The line 602 is empty.
The name of the author for the reference "Malkiel, B. G. (1973). A random walk down wall street [by] burton G. Malkiel. Norton." from the line 654 should be corrected: "Burton".
The references "Nofsinger, J. R. (2010). The Psychology of Investing, 4. bs., yy." from the line 657 is incomplete. Please revise.
Best regards!
Round 2
Reviewer 1 Report
Overall authors have tried to address all the points raised in the reviewer's comments.
In line with my earlier comment -Authors should make a proper table to highlight unexplored areas under the broad theme selected to give proper recommendations.
Authors have requested below clarification-
However, we would appreciate further clarification on what you mean by a "proper" table to highlight unexplored areas. Are you suggesting a table that lists specific areas for further research or a table that provides an overview of the existing literature and identifies gaps in our understanding? We would be grateful for any further feedback on this matter.
Authors could make a table to highlight the areas in which future research could be explored under the same broad theme.
This will strengthen the contribution of the paper.
Reviewer 2 Report
Dear Authors,
I have read the second version of your manuscript, but it still needs major corrections, because you didn't addressed all my recommendations from the previous round of review.
Please carefully address them one by one.
Athough you defined the genreral research questions in the Introduction, you should also define at least one research hypothesis to be tested. A scientific article must test research hypotheses and the readers needs to see the result of testing the RH.
At page 4, you included the figure with the steps of the methodology. Please include the title and the number of the figure.
At the line 249 you have figure 2. But before that, you have many figures and charts in the previous pages. Please revise and correct the numbers and the titles of them.
Starting with the line 410, there are some extra spaces between paragraphs. Please remove these space (see the lines 412 - 413, 426 - 427, 440 - 441, etc.).
As I already told you in the previous stage of review, you should improve the context of your research by including the following resources: https://doi.org/10.3390/math8010120, http://www.ecoforumjournal.ro/index.php/eco/article/view/884, https://doi.org/10.3390/joitmc6030071, https://doi.org/10.3390/jrfm15080319. I have read the second version of the article, but you didn't include them. I recommend you to use these references in your work.
The section "3. Results" and "3.1 Current status of the field" starts without any descriptive text. Please avoid this approach. Each new (sub)section should start with a short introductice sentence, so that the readers understand the context of that section.
The last part of the chart from the lines 189 - 190 is not relevant: you have partial data from 2023 (which should be eliminated from the chart) and the years 2024 and 2025 with empty values. I recommend you to remove the last 3 values from the chart.
In my version of the pdf document, the figure from the line 247 appears twice (the first figure is the old wrong figure from the previous version of the manuscript, and under it I can see the corrected version). Please revise.
In the final References list, there are some empty lines: 711, 725. Please revise them.
Best regards!
Round 3
Reviewer 2 Report
Dear Authors, please see the following remarks:
- the figure 3 from page 8 still appears twice in my version of the pdf document. The first version is the wrong one, with percentages greater than 100%. I don't know if it is because the Track&Trace option from Word editor. Please check this issue.
- figure 4 from page 9 presents the same data in the map and in the table. Please avoid presenting the same data twice. I recommend you to remove one of them.
- the figure from page 10 seems to not have a title. Please check this issue.
- at the line 310 you have now figure 1, but you also have many figures before.
- the lines 399 - 401 are empty. Please remove them.
- the font from figure 7 is very small.
- the final references list should be improved with "fresh" references. Now, most of them are very old.
Best regards!
